# Efficient GPT-4V level multimodal large language model for deployment on edge devices

Yuan Yao[1,2,3], Tianyu Yu[1], Ao Zhang[3], Chongyi Wang[4], Junbo Cui[4], Hongji Zhu[4], Tianchi Cai[4], Chi Chen[1], Haoyu Li[1], Weilin Zhao[1], Zhihui He[1], Qianyu Chen[5], Ronghua Zhou[4], Zhensheng Zou[4], Haoye Zhang[1], Shengding Hu[1], Zhi Zheng[4], Jie Zhou[4], Jie Cai[4], Xu Han[1], Guoyang Zeng[4], Dahai Li[4], Zhiyuan Liu ⬭[1] ✉ & Maosong Sun ⬭[1] ✉

Multimodal large language models have revolutionized AI research and industry, paving the way toward the next milestone. However, their large sizes and high computational costs restrict deployment to cloud servers, limiting use in mobile, offline, energy-sensitive, or privacy-critical scenarios. We present MiniCPM-V, efficient models for edge devices that integrate advancements in architecture, training, and data. The 8B model outperforms GPT-4V, Gemini Pro, and Claude 3 across 11 public benchmarks, processes high-resolution images at any aspect ratio, achieves robust optical character recognition, exhibits low hallucination rates, and supports over 30 languages while running efficiently on mobile phones. This progress reflects a broader trend: The sizes for high-performing models are rapidly decreasing alongside growing edge computation capacity, enabling advanced multimodal models to operate locally on consumer hardware. Such developments unlock applications across diverse real-world scenarios, from enhanced mobile AI to privacy-preserving solutions, marking a critical step toward democratizing powerful multimodal intelligence.

The rapid development of Multimodal Large Language Models (MLLMs)[1-11] has brought an impressive surge in multimodal capabilities in understanding, reasoning and interaction. This has not only fundamentally reshaped the landscape of AI research and industry, but also shed light on a promising path towards the next AI milestone. However, current MLLMs are still far from being practical in real-world applications. One of the most predominant challenges is the heavy computational burdens imposed by the massive number of parameters of MLLMs. As a result, most MLLMs can only be deployed on high-performing cloud servers, leading to significant energy consumption and carbon emissions. This limitation significantly constrains potential application scopes such as on mobile devices, energy-sensitive scenarios, offline settings without stable network connections, and privacy/security protective scenarios for both personal and industrial users.

In light of these limitations, there is a growing interest in exploring more efficient lightweight MLLMs[1,3,11,12] that can run on edge devices. Edge scenarios encompass a broader scope of equipment, including mobile phones, personal computers, vehicles and robotics, etc., which are ubiquitous in users' daily lives and are experiencing rapid advancements in computation capacities. On-device MLLMs provide a promising solution towards more practical applications due to their broader usage scope, better computation efficiency, more robust offline behaviors, and better privacy/security protection.

However, developing capable on-device MLLMs is challenging due to significantly constrained parameter and inference computation

[1]Tsinghua University, Beijing, China. [2]Shanghai Qi Zhi Institute, Shanghai, China. [3]National University of Singapore, Singapore, Singapore. [4]ModelBest Inc., Beijing, China. [5]The Chinese University of Hong Kong, Hong Kong, China. ✉e-mail: liuzy@tsinghua.edu.cn; sms@tsinghua.edu.cn

budgets. As a result, more careful architecture designs and training recipes are required to fully unleash the potential of on-device MLLMs. In this work, we present MiniCPM-V, a series of efficient MLLMs deployable on edge devices. The philosophy of MiniCPM-V is to achieve a good balance between performance and efficiency, a more important objective in real-world applications. From February to May in 2024, we unveiled three models: (1) In February, we launched MiniCPM-V 1.0 2B, an early prototype of on-device MLLMs. (2) In April, we released MiniCPM-V 2.0 2B, which outperforms strong larger MLLMs such as Qwen-VL 9B[7], CogVLM 17B[5], and Yi-VL 34B[13]. This iteration also introduces support for high-resolution image perception and exhibits promising OCR capabilities. (3) Most recently, in May, we introduced MiniCPM-Llama3-V 2.5 8B, which outperforms proprietary GPT-4V-1106, Gemini Pro and Claude 3 on the OpenCompass evaluation. Noteworthy features of this model include strong OCR capability, high-resolution image perception, trustworthy behavior, multilingual support, and efficient edge deployment optimization. The capabilities of on-device MLLMs have been growing even stronger in our later releases since May 2024.

More importantly, MiniCPM-V can be viewed as a representative example of a promising miniaturization trend of MLLMs. Figure 1 summarizes the recent development of MLLMs[3,12,14] in terms of performance, parameters and release time. We observe an interesting trend akin to Moore's Law[15] indicated by the red line: the sizes of models reaching GPT-4V level performance are rapidly decreasing over time. This phenomenon could perhaps be called Moore's Law of MLLMs. Simultaneously, the computational capacity of edge devices such as phones and personal computers is steadily increasing (qualitatively depicted by the blue line). The convergence of these two trends indicates usable (e.g., GPT-4V level) MLLMs deployable on edge devices are soon within reach, opening up broader possibilities

and benefiting more application scenarios in the near future. From a historical perspective of human technology development, this trend can also be viewed as the human pursuit of miniaturization of state-of-the-art technologies, which has been repeatedly witnessed in other science and technology fields. For example, in aerospace, the latest SpaceX Raptor 2 rocket engine can achieve a strong thrust of 2,256 kN with a mass of 1.6 tons, whereas 20 years ago, the RD-0750 rocket engine could only achieve a thrust of 1,413 kN with a mass exceeding 4 tons[16].

## MiniCPM-V Series Techniques

In this paper, we will take MiniCPM-Llama3-V 2.5 as an example, and systematically introduce the notable features of MiniCPM-V series and the key techniques behind them:

- *Leading Performance.* MiniCPM-Llama3-V 2.5 achieves better performance than GPT-4V-1106, Gemini Pro and Claude 3 on OpenCompass collection, a comprehensive evaluation over 11 popular benchmarks. This is jointly contributed by its careful design in architecture, data and training recipes, which we will detail in the following.
- *Strong OCR Capability.* MiniCPM-Llama3-V 2.5 outperforms GPT-4V, Gemini Pro and Qwen-VL-Max on OCRBench. It also supports high-utility functions such as table-to-markdown conversion and full OCR content transcription. These are largely attributed to the 1.8M pixel high-resolution (e.g., 1344 × 1344) image perception technique across any aspect ratios[17] of MiniCPM-Llama3-V 2.5.
- *Trustworthy Behavior.* Based on the RLAIF-V[18] and RLHF-V[19] techniques that align MLLM behaviors from high-quality AI/human feedback, MiniCPM-Llama3-V 2.5 exhibits more trustworthy behaviors, achieving lower hallucination rates than GPT-4V-1106 on Object HalBench.

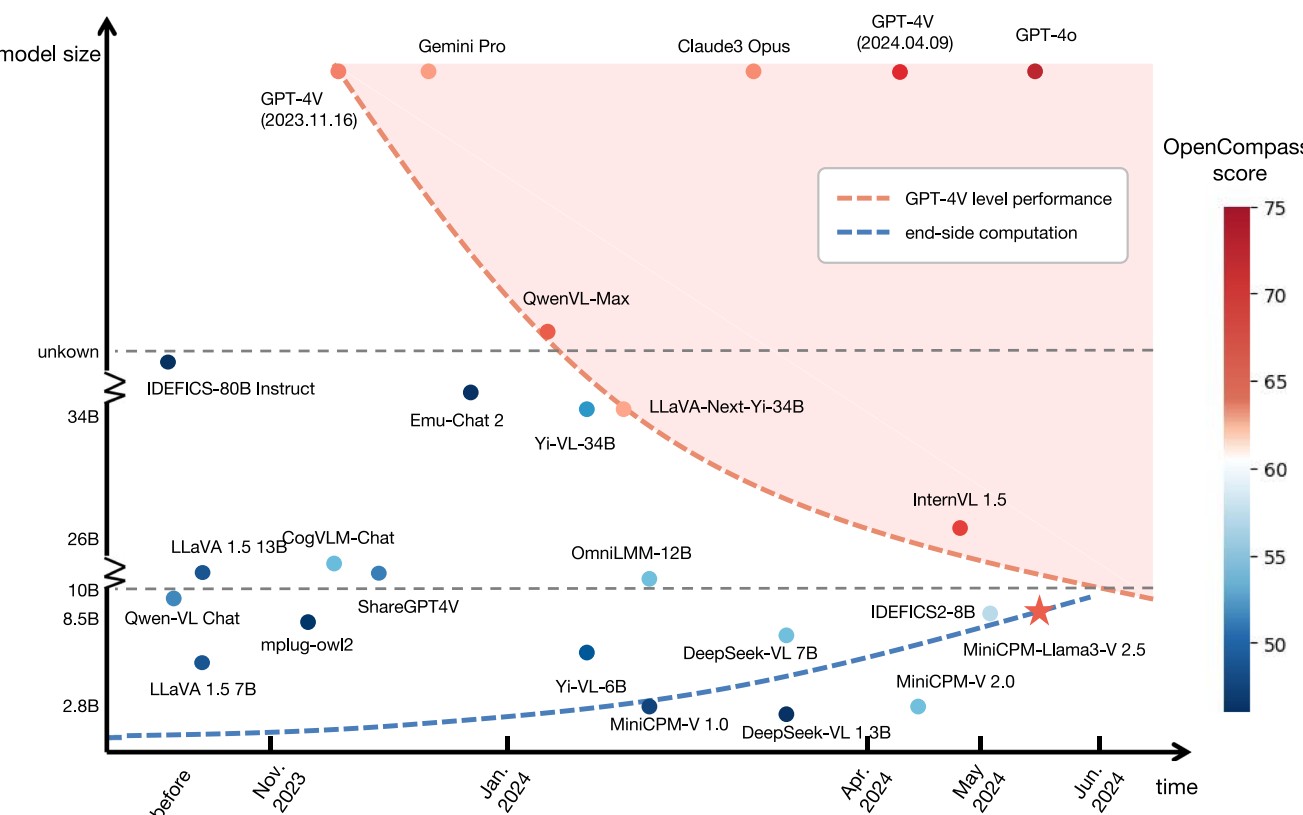

**Fig. 1 | Moore's Law for MLLM? Trends of MLLM development in terms of time (x-axis), model size (y-axis), and performance (color).** The red line shows the decreasing model sizes for achieving GPT-4V level performance, while the blue line represents the growing edge device computation capacity. This jointly shows that GPT-4V level MLLMs deployed on edge devices are becoming increasingly possible, unlocking a wider spectrum of real-world AI applications in the near future.

- *Multilingual Support.* Inspired by the findings from VisCPM[8], the integration of multilingual LLM significantly alleviates the heavy reliance on multimodal training data in low-resource languages. Based on the foundation, a lightweight multilingual multimodal instruction tuning helps MiniCPM-Llama3-V 2.5 generalize its multimodal capabilities to more than 30 languages.
- *Efficient Edge Deployment.* We systematically integrate a suite of on-device optimization techniques, encompassing quantization, memory optimization, compilation optimization and NPU acceleration, enabling efficient deployment on edge devices.

We hope MiniCPM-V series can serve as an example for unveiling the potential of on-device MLLMs, and help draw more attention to promote the research in this direction. Following Moore's Law for MLLM, we believe there will be increasingly powerful on-device MLLMs

with reduced sizes, bringing efficient, safe, and trustworthy AI services on devices soon.

## Results

### Overview of MiniCPM-V

As shown in Fig. 2b, MiniCPM-V comprises three key modules: the visual encoder, compression layer, and LLM. The input image is first encoded by a visual encoder, utilizing the adaptive visual encoding approach. The visual tokens are then compressed by the compression layer, which adopts a perceiver resampler structure with one layer cross-attention. Finally, the compressed visual tokens, along with the text input, are fed into the LLM for conditional text generation.

Encoding high-resolution images poses two major challenges. In terms of efficiency, directly encoding high-resolution images results in an excessive number of visual tokens, rendering it computationally

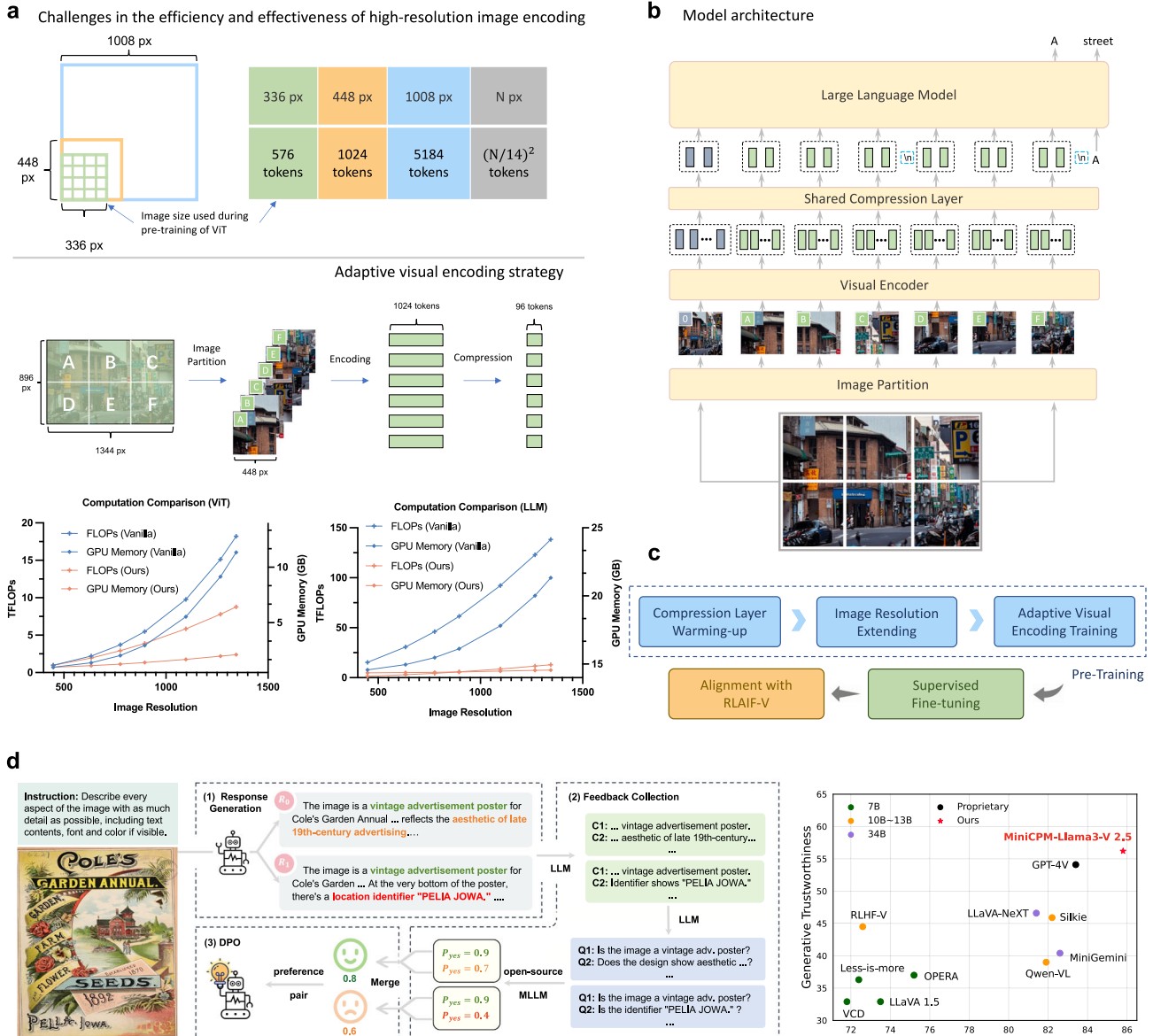

**Fig. 2 | Overview of MiniCPM-V. a** Conventional visual encoding requires a large number of tokens when encoding high-resolution images. Our proposed adaptive visual encoding strategy employs adaptive image partitioning and compressed encoding, significantly reducing computational costs when processing high-resolution images. **b** Overall structure presents the architecture of the model including the visual encoder, shared compression layer, and LLM. **c** The progressive multimodal learning strategy is applied to train MiniCPM-V, encompassing three phases: the pre-training phase, supervised fine-tuning phase, and alignment phase. **d** RLAIF-V framework for hallucination reduction. (1) Response generation produces multiple responses for an instruction using the policy model. (2) Feedback collection evaluates the correctness of each response in a divide-and-conquer fashion. (3) DPO optimizes the model on the preference dataset.

prohibitive for edge devices. In terms of effectiveness, the considerable discrepancy between the image resolution and the resolution employed during ViT pre-training can lead to out-of-distribution problems and therefore substantially degrade encoding performance. To address the challenges, we take advantage of the adaptive visual encoding strategy[17] as shown in Fig. 2a. To handle the high-resolution images with different aspect ratios, we divide images into slices, where each slice better matches ViT's pre-training setting in terms of resolution and aspect ratio. Each image is divided into a maximum of 10 slices, supporting 1.8 million pixels (e.g., 1344 × 1344 resolution) at most in total during encoding, which covers most real-world application scenarios. Then we adjust each slice by resizing it proportionally so that the resultant area size matches ViT pre-training area size, and interpolate the ViT's position embeddings to adapt to the slice's aspect ratio. After visual encoding, each slice is encoded into 1,024 tokens, where 10 slices can yield over 10k tokens collectively. To manage this high token count, we employ a compression module comprising of one-layer cross-attention and a moderate number of queries, with 2D positions informed[7]. In practice, the visual tokens of each slice are compressed into 64 queries for MiniCPM V1&2 and 96 tokens for MiniCPM-Llama3-V 2.5 through this layer. Compared with other MLLMs with competitive performance, the significantly smaller number of visual tokens in MiniCPM-V series enables superior efficiency in terms of GPU memory consumption, inference speed, first-token latency and power consumption, making it more friendly to wider application scopes and communities. Finally, we introduce a spatial schema inspired by[20] to indicate each slice's position relative to the whole image. We first wrap tokens of each slice by two special tokens "<slice>" and "<\slice>", and then employ a special token "\n" to separate slices from different rows.

We adopt a three-phase progressive multimodal learning strategy as shown in Fig. 2c, which consists of the pre-training phase, supervised fine-tuning phase, and alignment phase. In the first phase, we utilize large-scale image-text pairs for MLLM pre-training to align the visual modules (i.e., visual encoder and compression layer) with the input space of LLM and to acquire foundational multimodal knowledge. The pre-training phase can be further divided into three stages. In the first stage, the compression layer is warmed up. The second stage involves extending the input resolution of the pre-trained visual encoder. Finally, in the third stage, the visual modules are trained using an adaptive visual encoding strategy, allowing them to effectively handle high-resolution inputs with any aspect ratios. In the second phase, we perform Supervised Fine-Tuning (SFT) on high-quality visual question answering datasets to further learn knowledge and interaction capability from human annotations. We unlock all model parameters to better exploit the data and learn rich knowledge during the SFT phase. We also conduct a lightweight yet high-quality SFT process as in VisCPM[8] to enhance alignment with languages beyond English and Chinese, achieving strong multimodal performance across more than 30 languages. In the alignment phase, we employ the recent RLAIF-V[18] approach to address the hallucination problem (Fig. 2d), where the MLLM generates responses that are not factually grounded in the input image[19]. The first step of RLAIF-V is to generate multiple responses for a given instruction using the policy model. Then a divide-and-conquer strategy is applied for response scoring. After collecting the high-quality AI feedback, we perform preference learning via DPO[21] method.

This paper introduces the first 3 models in the MiniCPM-V series, including MiniCPM-V 1.0, MiniCPM-V 2.0, and MiniCPM-Llama3-V 2.5. MiniCPM-V 1.0 is trained with the pre-training stage1&2 and SFT without using the adaptive visual encoding and RLAIF-V. For MiniCPM-V 2.0, we include all training stages and the adaptive visual encoding strategy to further improve performance. MiniCPM-Llama3-V 2.5 adopts Llama3-Instruct 8B as its base LLM, showcasing strong multimodal understanding capabilities, as illustrated in Fig. 3.

## Evaluation across diverse multimodal understanding benchmarks

We perform a comprehensive evaluation on popular benchmarks covering visual question answering, multimodal conversation, knowledge and reasoning, OCR, and hallucination. (1) *General benchmarks*. We adopt OpenCompass[22] as the general evaluation indicator, which is a comprehensive collection over 11 popular multimodal benchmarks, including MME[23], MMBench[24], MMMU[25], MathVista[26], LLaVA Bench[4], etc. We also report the results on RealWorldQA for real-world spatial understanding capabilities. (2) *OCR benchmarks*. We adopt three widely used benchmarks for OCR capability evaluation, including including OCRBench[27], TextVQA[28] and DocVQA[29]. (3) *Hallucination benchmarks*. We also include Object HalBench[19,30] to evaluate the trustworthiness of the models.

We compare with strong baselines in different series: For open-source models, we compare with strong models including Yi-VL-6B/34B[13], Qwen-VL-Chat-9B[7], DeepSeek-VL-7B[3], TextMonkey[31], CogVLM-Chat-17B[5], CogVLM2-Llama3-19B[5], Idefics2-8B[9], Bunny-Llama-3-8B[32], XTuner-Llama-3-8B-v1.1[33], LLaVA-NeXT-Llama-3-8B[34], Cambrian-8B/34B[35], LLaVA-NeXT-Yi-34B[36], DeepSeek-VL-1.3B[3], MobileVLM V2[37], Mini-Gemini[14] and Phi-3-Vision-128k-instruct[12]. For proprietary models, we compare with GPT-4V-1106[2], Gemini-Pro[1] and Claude 3 Opus[38].

From the experimental results in Fig. 4, we have the following observations: (1) MiniCPM-Llama3-V 2.5 outperforms strong open-source models by a notable margin. For instance, MiniCPM-Llama3-V 2.5 surpasses the recent strong Idefics2-8B by 7.9 points on the OpenCompass benchmark, with similar model sizes. It also achieves better results than significantly larger models such as Cambrian-34B, LLaVA-NeXT-Yi-34B, Yi-VL-34B and CogVLM2-Llama3-19B. (2) Compared with powerful proprietary models, such as GPT-4V-1106 and Gemini Pro, MiniCPM-Llama3-V 2.5 achieves better performance on the OpenCompass benchmark with significantly fewer parameters. In addition, MiniCPM-Llama3-V 2.5 also achieves lower hallucination rates than GPT-4V-1106 on Object HalBench, indicating its trustworthiness for real-world applications. (3) The smaller MiniCPM-V 2.0 with 2B parameters achieves significantly better performance compared with other 2B ~ 4B models, and is even comparable with 8B MLLMs such as Bunny-Llama-3-8B. In summary, the results show that MiniCPM-V series achieves a good balance between performance and efficiency, making it more friendly for broader communities and applications.

MiniCPM-V models also show strong OCR capabilities, including scene-text, document and screenshot understanding. As shown in Fig. 5a, MiniCPM-Llama3-V 2.5 outperforms open-source MLLMs ranging 1.7B–34B on OCRBench, TextVQA, and DocVQA. The performance on these datasets is even comparable to proprietary models like GPT-4V-1106 and Gemini Pro. MiniCPM-V 2.0 also achieves significantly better performance among models in the 2B–4B parameter range (Fig. 5b).

Based on the multilingual multimodal generalization approach from VisCPM, MiniCPM-Llama3-V 2.5 extends its multimodal capability to over 30 languages. As shown in Fig. 5c, MiniCPM-Llama3-V 2.5 can outperform Yi-VL 34B and Phi-3-vision-128k-instruct on the multilingual LLaVA Bench. The promising multilingual multimodal capability makes MiniCPM-Llama3-V 2.5 useful in serving larger linguistic groups.

To investigate the effectiveness of key components, we perform an ablation study on high-resolution perception and multi-stage training pipeline. To ablate high-resolution perception, we follow the standard method in LLaVA-1.5[39] to downsample high-resolution images into low-resolution versions (i.e., 448 × 448) in both training and inference. To ablate multi-stage training pipeline, we follow the standard two-stage training (i.e., pretraining and instruction tuning) in LLaVA-1.5[39]. Due to the high computational costs of model training on full data, we perform ablation on a subset of full training data by randomly sampling 10% data from each dataset, resulting in 70M data in

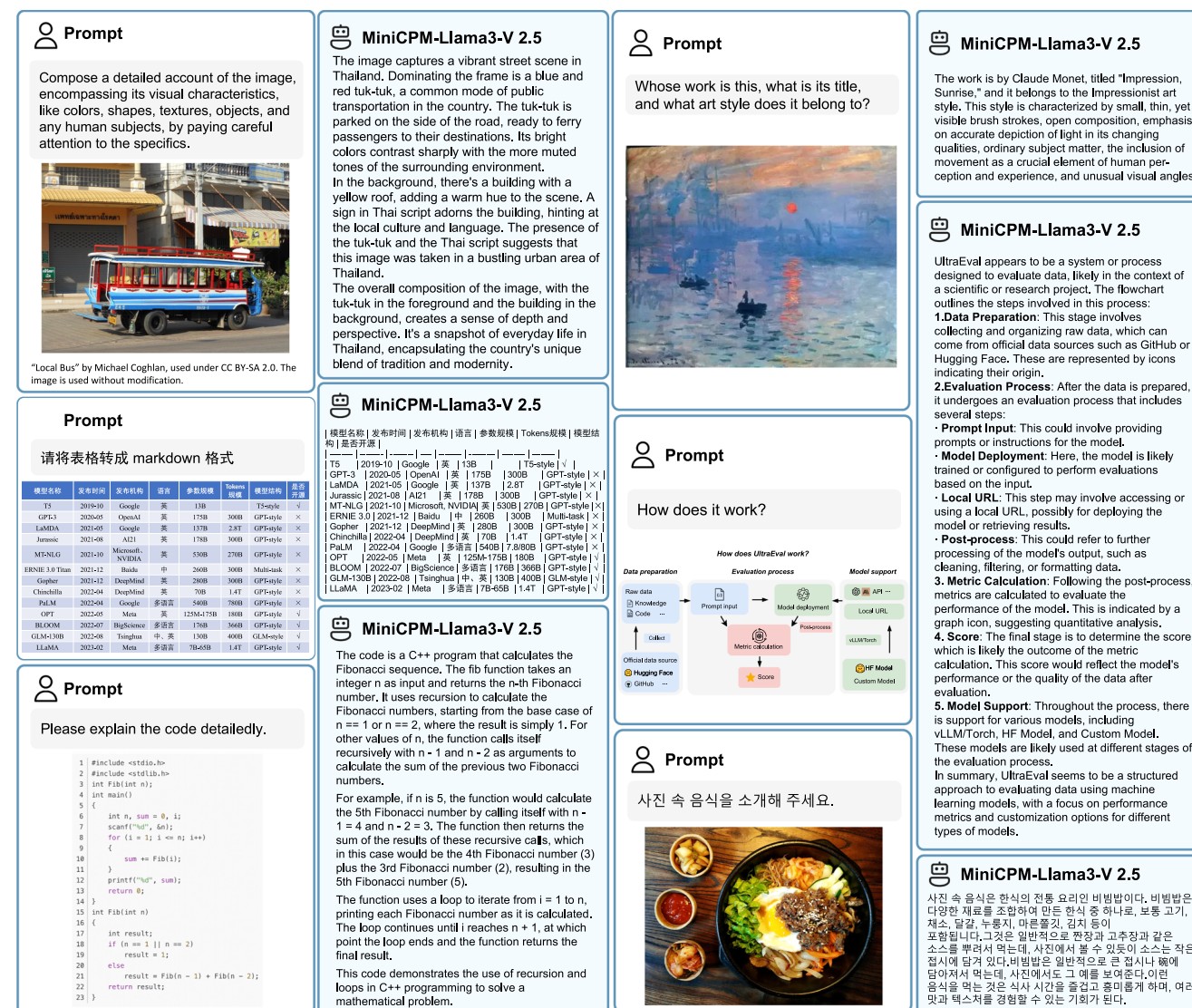

**Fig. 3 | Qualitative results of MiniCPM-Llama3-V 2.5.** MiniCPM-Llama3-V 2.5 demonstrates strong performance in a variety of tasks, including text recognition in images, table-to-markdown conversion, complex reasoning, and multilingual interactions.

total, which is sufficient to validate a frontier MLLM. From the experimental results in Table 1, we can see that both high-resolution perception and multi-stage training pipeline contribute to the final performance. The reason is that high-resolution perception is crucial for MLLMs to perceive fine-grained visual details, especially for OCR-related tasks, and multi-stage training can better fit and exploit training data of different forms and qualities.

Moreover, it is worth noting that MiniCPM-Llama3-V 2.5 requires significantly less inference computation. For example, the visual token number range of MiniCPM-Llama3-V 2.5 is (96, 960), which is lower than LLaVA-NeXT-Llama-3-8B's (1728, 2880). This can be important especially for real-world on-device applications in terms of inference speed, first-token latency, memory usage, and power consumption.

Specifically, we provide a comparison of the computational costs between the adaptive visual encoding method and the vanilla visual encoding method (i.e., visual features of the original image from ViT are directly projected and input into the LLM, as in LLaVA-1.5[39]). From the results in Fig. 2a, we can see that compared with the standard method, adaptive visual encoding largely reduces both FLOPs and GPU memory usage in both ViT and LLM for high-resolution images. The reason is that, compared with the standard method, image slicing

prevents the quadratic computation growth of ViT, and the compression layer largely reduces the number of visual tokens to LLMs.

### Efficient Deployment of MiniCPM-V on Edge Devices

In this section, we investigate the deployment of MiniCPM-V on edge devices. Edge devices such as smartphones and computers often face resource constraints due to factors like heat dissipation, size limitations, and power consumption. When deploying models, the two most critical limitations are memory capacity and CPU/GPU processing speed. High-performance servers typically boast extensive memory capacities, often exceeding 100GB or even 1TB. In contrast, the memory available on mobile phones typically ranges from 12GB to 16GB, which can be insufficient for MLLM deployment. On the other hand, The overall processing speeds of CPUs in smartphones are notably slower. For instance, the Snapdragon 8 Gen3 features 8 CPU cores, whereas high-performance server like Intel Xeon Platinum 8580 has 60 CPU cores. Similarly, mobile phone GPUs are not as powerful as server GPUs. For example, Qualcomm Adreno 750 only has 6 TFLOPS, while NVIDIA 4090 can reach 83 TFLOPS.

To deploy the MLLM on edge devices, we first employ quantization for reduced memory cost. For MiniCPM-Llama3-V 2.5, the fp16

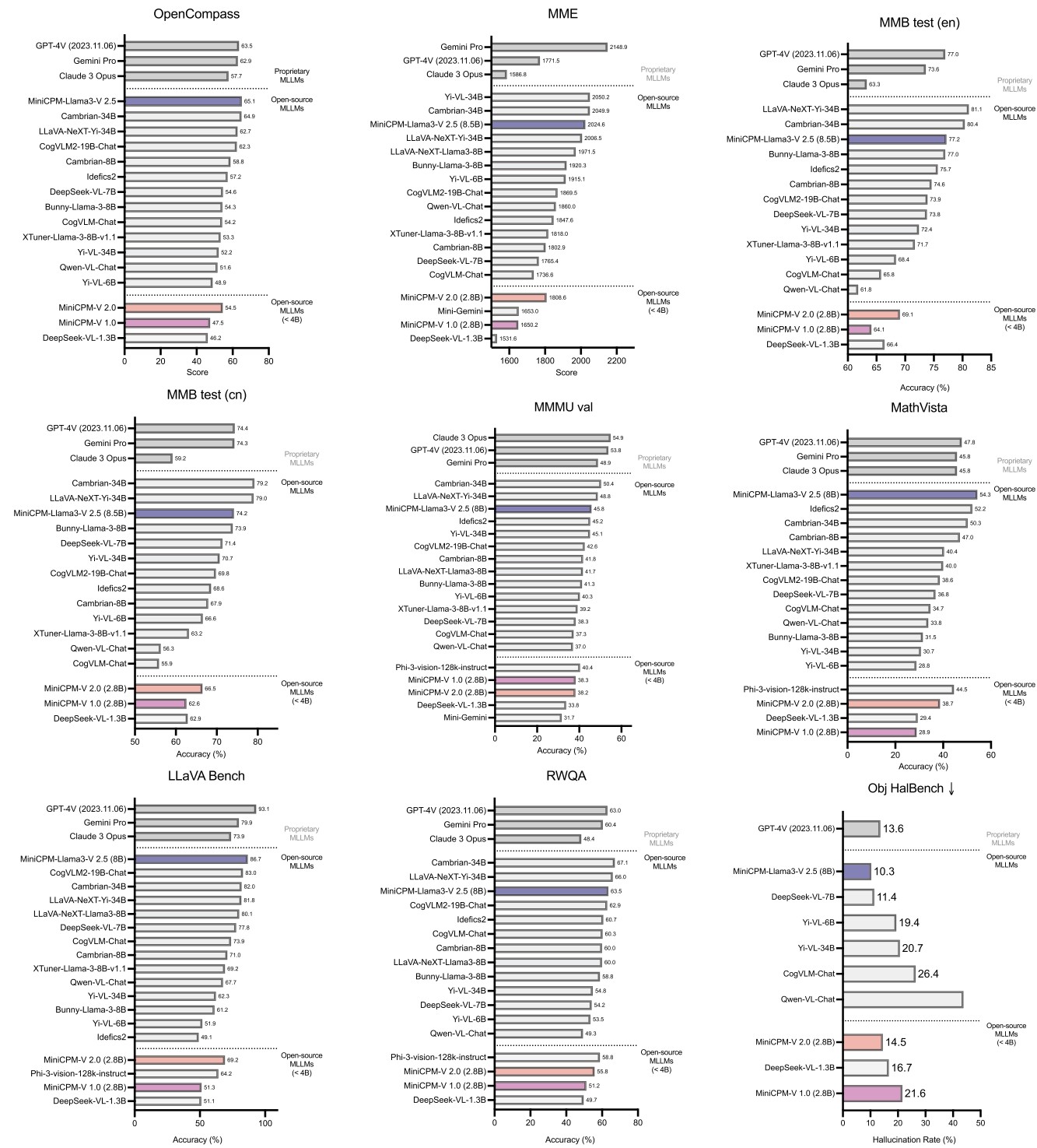

**Fig. 4 | Performance comparison of proprietary MLLMs, open-source MLLMs on general multimodal benchmarks.** MiniCPM-Llama-V 2.5, with only 8 billion parameters, outperforms leading open-source MLLMs and achieves superior results on the OpenCompass benchmark compared to proprietary models like GPT-4V-1106 and Gemini Pro. In addition, MiniCPM-V 2.0, with 2 billion parameters, significantly outperforms other MLLMs with fewer than 4 billion parameters.

version model typically demands 16–17G memory. We opt for the Q4_K_M mode 4-bit quantization strategy within GGML framework. This reduces the memory requirement to around 5G, which is friendly to mobile phone usage. We then empirically investigate the deployment results on different frameworks. Several frameworks have been proposed for on-device deployment. Illustrated in Fig. 6, we make a thorough investigation of different frameworks for different chip types including CPU, GPU, and NPU. Given the ubiquity of CPU usage

across devices, we prioritize this chip type and opt for the llama.cpp[40] framework. Combining quantization and llama.cpp on Xiaomi 14 Pro (Snapdragon 8 Gen 3), the model achieves a text encoding latency of 64.2s and a text decoding speed of 1.3 tokens/s (as depicted in Fig. 6f), which is still far from acceptable for users.

To achieve better acceleration, we further investigate a series of advanced techniques including memory usage optimization, compilation optimization, configuration optimization, and NPU acceleration,

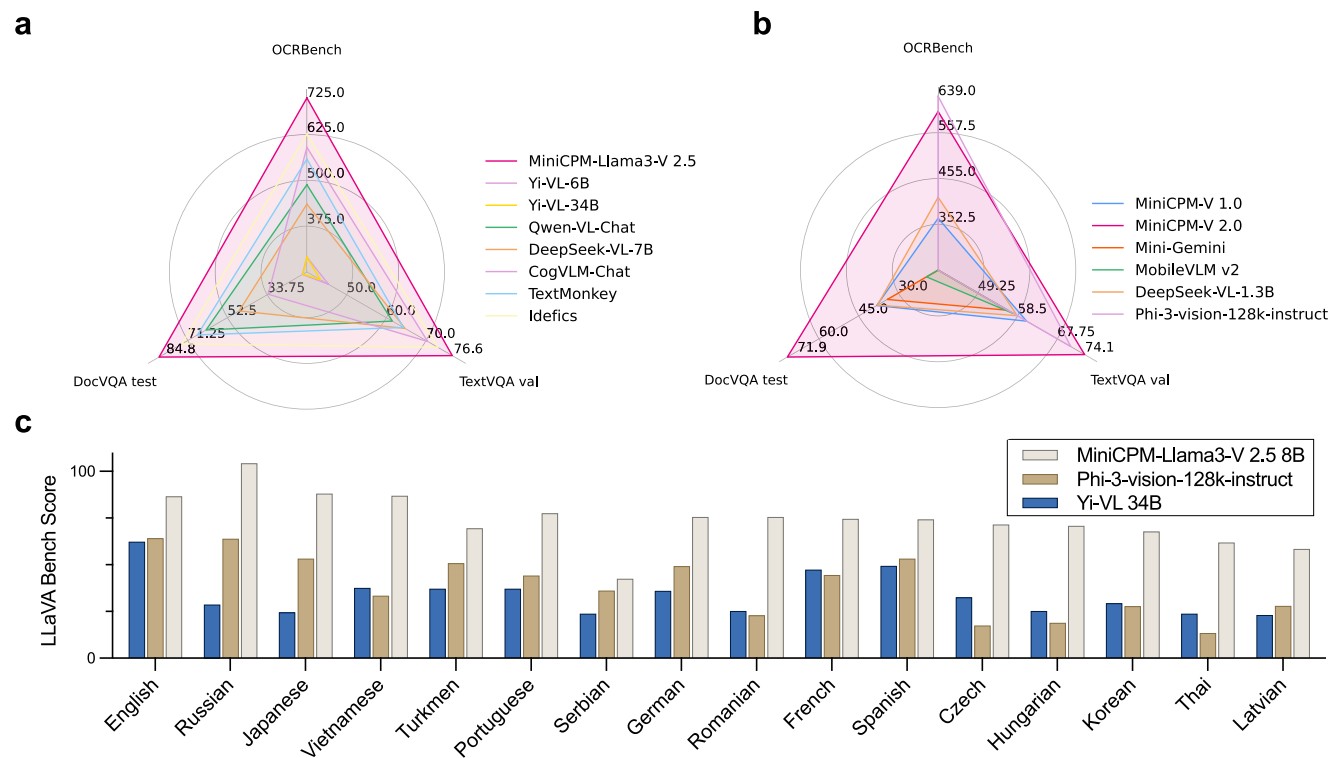

**Fig. 5 | Performance on OCR and multilingual benchmarks. a, b** Results on OCR benchmarks including OCRBench, the DocVQA test set and the TextVQA val set for (**a**) open-source MLLMs (>4B) and (**b**) MLLMs (<4B). **c**, Multilingual multimodal capabilities on the multilingual LLaVABench.

**Table 1 | Multi-stage training and high-resolution perception ablation results on a subset (10%) of full training data**

| Model | OpenCompass 2.0 | MMMU (val) | MMVet | MMBv1.1 EN (val) | OCRBench | ChartQA (test) | MME | TextVQA (val) | DocVQA (test) |
|---|---|---|---|---|---|---|---|---|---|
| MiniCPM-Llama3-V 2.5 | **54.1** | **43.3** | **46.7** | **69.6** | **697** | **65.4** | **1922** | **72.5** | **79.5** |
| w/o multi-stage training | 50.1 | 42.9 | 44.9 | 67.5 | 549 | 57.6 | 1544 | 70.1 | 64.7 |
| w/o high-resolution perception | 48.5 | 37.9 | 37.8 | 64.2 | 580 | 51.2 | 1701 | 65.1 | 58.4 |

Bold font indicates the best result on each test set.

as shown in Fig. 6a–d. We first explore memory usage optimization strategies to address the image processing bottleneck of the inference speed due to limited memory resources on mobile phones. Instead of loading both ViT and LLM simultaneously into memory, we adopt a sequential loading approach. Specifically, we first load ViT for visual encoding, followed by the LLM for visual and text token encoding. By releasing the large amount of memory occupied by LLM, we can prevent frequent paging (swapping in and out) during ViT encoding, thereby improving the program efficiency. This optimization technique, as illustrated in Fig. 6e, results in a notable reduction of image processing time from 45.2s to 31.5s. We also find that directly compiling the models on the target devices can significantly improve the encoding latency and the decoding throughput. This can be attributed to better consistency between the compilation and target device instruction set architecture. As depicted in Fig. 6e, this optimization endeavor yields promising results. Encoding latency shows a notable reduction from 50.5s to 17.0s, while decoding throughput experiences a significant boost from 1.3 tokens/s to 3.2 tokens/s. Next, instead of relying on a single default configuration of the llama.cpp framework, we propose an automated parameter search algorithm that dynamically identifies the optimal configurations (e.g., computational distribution across CPU cores) tailored to various edge devices. Through configuration optimization, we can achieve good improvements. Specifically, decoding throughput surged from 3.2 tokens/s to an impressive 8.2 tokens/s, surpassing the typical human reading speed. Finally, we leverage Neural Processing Units (NPUs), a class of specialized hardware designed to accelerate AI applications, available in certain smartphones. Recognized for their ability to address computational bottlenecks, NPUs enable accelerated visual encoding. Specifically, we replace the backend framework of ViT with QNN while retaining the llama.cpp backend for the language model component. On mobile devices equipped with Qualcomm NPUs, this optimization yields a notable reduction in visual encoding time, decreasing from 3.7 to 1.3 s.

For a comprehensive assessment of MiniCPM-Llama3-V 2.5's performance across various edge devices, we present test results on Xiaomi 14 Pro (Snapdragon 8 Gen 3), vivo X00 Pro (Mediatek Dimensity 9300), Macbook Pro (M1), and Jetson AGX Orin 32G in Fig. 6f. Thanks to the deployment optimization techniques, MiniCPM-Llama3-V 2.5 can operate efficiently on both mobile phones and personal computers, delivering acceptable latency and throughput. For instance, leveraging NPU on Xiaomi 14 Pro enables it to achieve a similar encoding speed as the Mac M1. Furthermore, nearly all devices exhibit comparable or higher throughput compared with human reading speed. Upon analyzing the results, it becomes evident that the current computation bottleneck primarily stems from LLM prefilling, which mainly involves encoding image and text tokens for LLM inference. Promising research directions involve developing more efficient

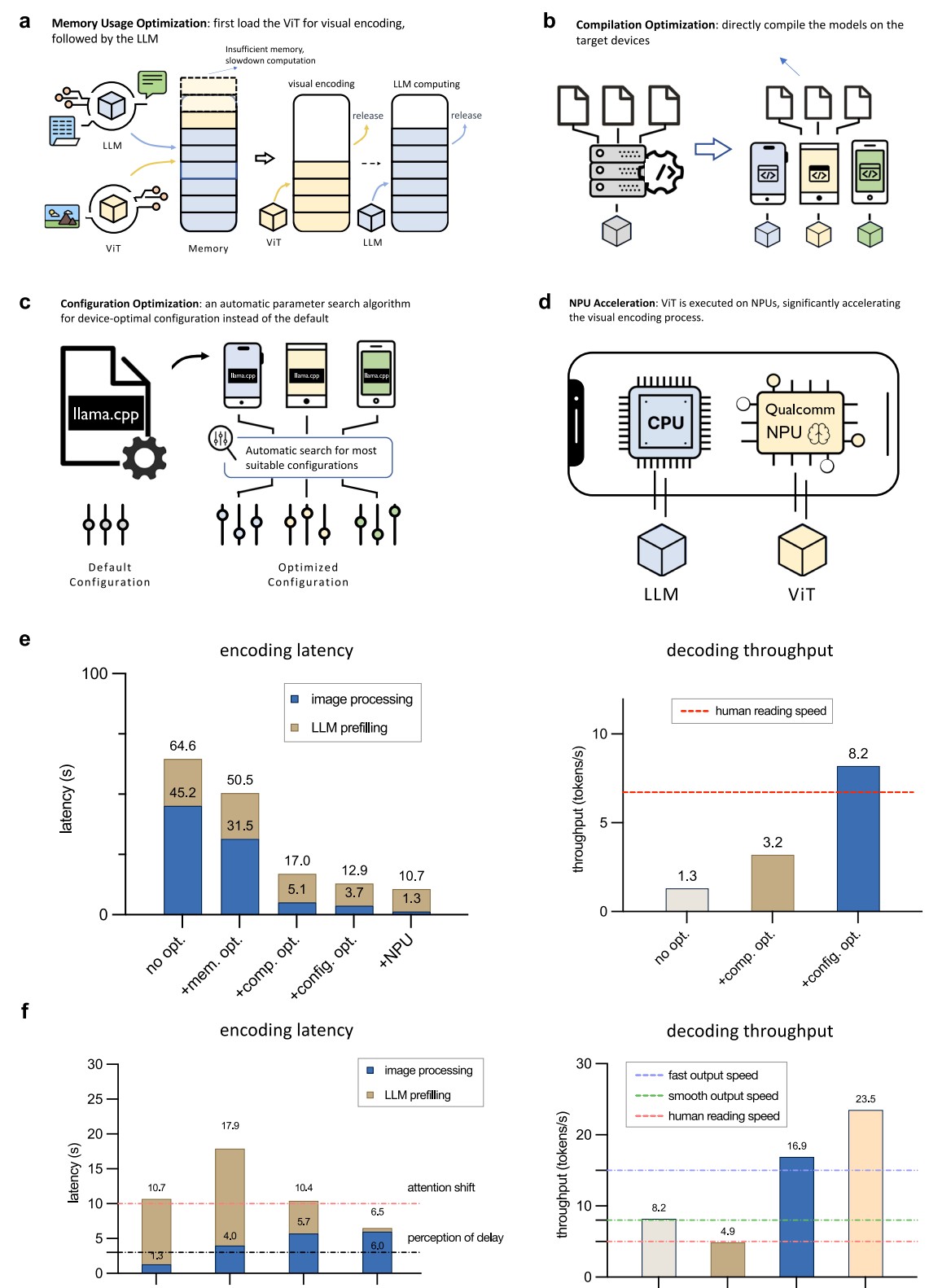

**Fig. 6 | Deploying MiniCPM-V on edge devices. a–d** Advanced techniques used in the deployment of MiniCPM-V on edge devices, including (**a**) memory usage optimization, (**b**) compilation optimization, (**c**) configuration optimization, and **d** NPU acceleration. **e** The influence of these techniques on the encoding latency and decoding throughput. The results are tested on the Xiaomi 14 Pro (Snapdragon 8 Gen 3). No opt.: non-optimized, mem. opt.: memory usage optimization, comp. opt.: compilation optimization, config. opt.: configuration optimization, NPU: NPU acceleration. **f** Results on different edge devices. We show the encoding latency and decoding throughput across different device types. Xiaomi 14 Pro is the only device with NPU.

visual encoding methods with fewer visual tokens, and better leveraging GPU/NPU acceleration for LLM encoding. With increasing attention to on-device MLLMs and the rapid advancement of GPU/NPU acceleration techniques, we believe that real-time interaction with on-device MLLMs can be reached soon.

## Discussion

The MiniCPM-V series models are a primary exploration into powerful on-device MLLMs. Thanks to techniques such as adaptive visual encoding, multilingual generalization, and the RLAIF-V method, MiniCPM-Llama3-V 2.5 can achieve GPT-4V level performance with significantly fewer parameters. Leveraging diverse optimization techniques for edge deployment, this model ensures an acceptable user experience on mobile phones.

Despite promising performance, there remain several limitations with the current MiniCPM-V models. (1) Capability Depth. There is still plenty of room for improvement in enhancing multimodal understanding capability and inference efficiency. (2) Capability Width. In addition to image modality, it's promising to expand MLLM capabilities to encompass other modalities, such as video and audio, etc., where GPT-4o[41] and Google Astra[42] have given good examples.

In addition to MLLM capabilities, edge deployment also presents unique challenges. The inference speed and latency are still far from good enough and the model service can be limited by the battery capacity. In addition, previous efforts on chips and deployment frameworks mainly target CNNs and LSTMs, which can be sub-optimal for MLLMs. Tailored efforts to MLLMs can bring ample room for improvement.

Considering the current limitations and the promising future of on-device MLLMs, we anticipate increasing efforts from both academia and industry in enhancing model capabilities in terms of depth and width, and improving smartphone chips and deployment frameworks. We believe that simultaneous advancements in model capability and edge device capacity will lead to on-device applications providing a satisfying user experience in the near future.

## Methods

### Adaptive visual encoding

**Image partition.** To process high-resolution images with varying aspect ratios, we divide them into slices[17]. Each slice is adjusted to more closely align with ViT's pre-training settings in terms of both resolution and aspect ratio. Specifically, we first calculate the ideal number of slices based on the input image size. Given an image with resolution $(W_I, H_I)$ and a ViT pre-trained on images with resolution $(W_v, H_v)$, we calculate the ideal slice number $N = \lceil \frac{W_I \times H_I}{W_v \times H_v} \rceil$. Then, we choose the combination of rows $n$ and columns $m$ from the set $\mathbb{C}_N = \{(m, n) | m \times n = N, m \in \mathbb{N}, n \in \mathbb{N}\}$. A good partition $(m, n)$ should result in slices that match well with ViT's pre-training setting. To achieve this, we use a score function to evaluate each potential partition:

$$S(m, n) = - \left| \log \frac{W_I/m}{H_I/n} - \log \frac{W_v}{H_v} \right|. \quad (1)$$

We select the partition with the highest score from all possible candidates:

$$m^*, n^* = \arg \max_{(m,n) \in \bar{\mathbb{C}}} S(m, n), \quad (2)$$

where $\bar{\mathbb{C}}$ is the possible $(m, n)$ combinations with the product $N$. However, when $N$ is a prime number, the feasible solutions can be limited to $(N, 1)$ and $(1, N)$. Therefore, we additionally introduce $\mathbb{C}_{N-1}$ and $\mathbb{C}_{N+1}$, and set $\bar{\mathbb{C}} = \mathbb{C}_{N-1} \cup \mathbb{C}_N \cup \mathbb{C}_{N+1}$. In practice, we set $N < 10$, supporting 1.8 million pixels (e.g., 1344 × 1344 resolution) at most during encoding. Although we can encompass more image slices for

higher resolutions, we purposely impose this resolution upper-bound, since it already well covers most real-world application scenarios, and the benefit of further increasing encoding resolution is marginal considering the performance and overhead.

**Slice encoding.** Although image partitioning can ensure a good match between the slices and the ViT pre-training setting, each slice's size is not precisely equal to $(W_v, H_v)$. To feed the slices into ViT, we first adjust each slice by resizing it proportionally so that the resultant area size matches ViT pre-training area size $W_v \times H_v$. This adjustment helps prevent a significant gap between the number of encoded patches and the ViT's pre-training setting. Subsequently, we interpolate the ViT's position embeddings to adapt to the slice's ratio. This involves reshaping the ViT's 1D embedding $\mathbf{P}_1 \in \mathbb{R}^{Q \times l}$ back to its 2D format $\mathbf{P}_2 \in \mathbb{R}^{q \times q \times l}$, where the number of position embeddings $Q = q \times q$. Then, we interpolate $\mathbf{P}_2$ to fit the size of each slice via 2D interpolation. We also include the original image as an additional slice to provide holistic information about the entire image.

**Token compression.** After visual encoding, each slice is encoded into 1,024 tokens, where 10 slices can yield over 10k tokens collectively. To manage this high token count, we employ a compression module comprising of one-layer cross-attention and a moderate number of queries, with 2D positions. In practice, the visual tokens of each slice are compressed into 64 queries for MiniCPM V1&2 and 96 tokens for MiniCPM-Llama3-V 2.5 through this layer. Compared with other MLLMs with competitive performance, the significantly smaller number of visual tokens in MiniCPM-V series enables superior efficiency in terms of GPU memory consumption, inference speed, first-token latency and power consumption, making it more friendly to wider application scopes and communities.

**Spatial schema.** To indicate each slice's position relative to the whole image, inspired by[20], we introduce a spatial schema. We first wrap tokens of each slice by two special tokens "<slice>" and "<\slice>", and then employ a special token "\n" to separate slices from different rows.

### Pre-training

In this phase, we utilize large-scale image-text pairs for MLLM pre-training. The primary goal of this phase is to align the visual modules (i.e., visual encoder and compression layer) with the input space of the LLM and learn foundational multimodal knowledge. We show the pre-training data composition in Table 2. The pre-training phase is further divided into 3 stages.

**Stage-1.** The role of stage-1 is to warm up the compression layer, primarily connecting the visual encoder and LLMs. (1) Trainable Modules. We randomly initialize the compression layer and train this module in stage-1, keeping other parameters frozen. The visual encoder's resolution is set to 224 × 224, which is the same as the visual encoder's pre-training setting. (2) Data. To warm up the compression layer, we randomly select 200M data from the Image Captioning data in Table 2. Data cleaning is performed to remove image-text pairs with poor correlation and ill-formatted text data, ensuring the data quality.

**Stage-2.** After the warm-up training of the compression layer, the role of stage-2 is to extend the input resolution of the pre-trained visual encoder. (1) Trainable Modules. In stage-2, we extend the image resolution from 224 × 224 to 448 × 448. The whole visual encoder is trained, leaving other parameters frozen. (2) Data. To extend the pre-trained resolution, we additionally select 200M data from the Image Captioning data in Table 2.

**Stage-3.** After extending the primary input resolution of the visual encoder, we finally train the visual modules using the adaptive visual

**Table 2 | Pre-training data**

| Category | | Sources | Size |
|---|---|---|---|
| Image Captioning | English | COCO[47], VG[48], CC3M[49], CC12M[50] | 410M |
| | | LAION-COCO[43], COYO[44], LAION-2B[43] | |
| | Chinese | AIC[51], LAION-2B-Chinese[43], WuKong[52] | 110M |
| | | Zero-Chinese[53], etc. | |
| OCR+Knowledge | English | WIT[54], IDL[55], SynthText[56], SynthDoG-en[57] | 39M |
| | | SynthDoG-zh[57], ArxivCap[58], etc. | |
| | Chinese | WIT[54], LAION-2B-OCR | 11M |

The pre-training data consists of image captioning and OCR data in English and Chinese. LAION-2B-OCR is generated by applying OCR tools to LAION-2B images.

**Table 3 | SFT data for MiniCPM-V series**

| Category | | Sources | Size |
|---|---|---|---|
| Part-1 | Short Caption | Flickr-30K[59], COCO[47] | 560K |
| | VQA | FM-IQA[60], VGQA[48], IconQA[61], GQA[62], VQAv2[63] | 1.4M |
| | | CLEVR[64], VizWiz[65], Visual7W[66], COCO-QA[67] | |
| | Knowledge | OKVQA[68], A-OKVQA[69], KVQA[70], ScienceQA[71] | 60K |
| | Grounding | RefCOCO[72] | 570K |
| | Reasoning | COMVINT[73], VCR[74], NLVR[75], LRV[76] | 135K |
| | Math | GeoQA[77], SMART-101[78] | 125K |
| | OCR | DocVQA[29], TextVQA[28], OCR-VQA[79], ST-VQA[80], VisualMRC[81], DVQA[82] | 1.7M |
| | | FigureQA[83], ChartQA[84], DeepForm[85], TabFact[86], InfographicsVQA[87] | |
| | | Kleister Charity[88], WikiTableQuestions[89], Real-CQA[90], AI2D[91], etc. | |
| | Chat | FSVQA[92], Visual-Dialog[93] | 780K |
| Part-2 | Part-1 | sample from Part-1 data | 400K |
| | OCR | DocVQA, TextVQA, OCR-VQA, VisualMRC, ChartQA, AI2D | 690K |
| | | ArxivQA[58], LLaVAR[94], TextOCR-GPT4V[95], etc. | |
| | Instruct | SVIT[96], LLaVA-Instruct-150K[4], UniMM-Chat[97], ShareGPT4V[98] | 1.9M |
| | | LVIS[99], ALLaVA[100] | |
| | Text-Only | UltraChat[101], Alpaca[102], ShareGPT[103], BELLE[104] | - |
| | | OpenOrca[105], OpenHermes[106], In-House-MiniCPM-SFT | |

Part-1&2 data are concatenated sequentially in the SFT phase. Part-1 focuses on bolstering basic recognition capabilities, while part-2 aims to enhance advanced capabilities in generating detailed responses and following human instructions.

encoding strategy, which can further accommodate high-resolution inputs with any aspect ratio. (1) Trainable Modules. During the stage-3 training, both the compression layer and the visual encoder are trained to adapt to the language model embedding space. The LLM is kept frozen to avoid disruption from the relatively low-quality pre-training data. (2) Data. Different from the previous stages with only image captioning data, during the high-resolution pre-training stage, we additionally introduce OCR data to enhance the visual encoders' OCR capability.

**Caption rewriting.** Image-text pairs sourced from the Web[43,44] can suffer from quality issues in the caption data, including non-fluent content, grammatical errors, and duplicated words. Such low-quality data can lead to unstable training dynamics. To address the issue, we introduce an auxiliary model for low-quality caption rewriting. The rewriting model takes the raw caption as input and is asked to convert it into a question-answer pair. The answer from this process is adopted as the updated caption. In practice, we leverage GPT-4[45] to annotate a small number of seed samples, which are then used to fine-tune an LLM for the rewriting task.

**Data packing.** Samples from different data sources usually have different lengths. The high variance of sample lengths across batches will

lead to inefficiency in memory usage and the risk of out-of-memory problem. To address the issue, we pack multiple samples into a single sequence with a fixed length. By truncating the last sample in the sequence, we ensure uniformity in sequence lengths, facilitating more consistent memory consumption and computational efficiency. Meanwhile, we modify the position ids and attention masks to avoid interference between different samples. In our experiments, the data packing strategy can bring 2 ~ 3 times acceleration in the pre-training phase.

**Multilingual generalization.** Multimodal capability across multiple languages is essential for serving users from broader communities. Traditional solutions involve extensive multimodal data collection and cleaning, and training for the target languages. Fortunately, recent findings[8] have shown that the multimodal capabilities can be efficiently generalized across languages via a strong multilingual LLM pivot, largely alleviating the heavy reliance on multimodal data in low-resource languages. In practice, we only pre-train our model on English and Chinese multimodal data, and then perform a lightweight but high-quality multilingual supervised fine-tuning to align to the target languages. Despite its simplicity, we find the resultant MiniCPM-Llama3-V 2.5 can achieve good performance in over 30 languages as compared with significantly larger MLLMs.

## Supervised fine-tuning

After learning foundational capabilities from pre-training, we perform supervised fine-tuning (SFT) on high-quality visual question answering datasets to further learn knowledge and interaction capability from human annotations.

**Trainable Modules.** Compared with the pre-training phase which mainly uses crawled data from the Web, the SFT phase mainly utilizes high-quality datasets annotated by either human lablers or strong models such as GPT-4. Therefore, we unlock all model parameters to better exploit the data and learn rich knowledge during SFT phase.

**Data.** Recent works[1,46] show that data near the end of training plays a more important role in shaping the models' capabilities and response styles. We categorize the SFT data into two parts, as shown in Table 3. Part-1 focuses on bolstering the models' basic recognition capabilities, while part-2 is tailored to enhance their capabilities in generating detailed responses and following human instructions. Specifically, part-1 data consists of the traditional question answering and captioning datasets with relatively short response lengths, which helps enhance the model's basic recognition capabilities. In comparison, part-2 encompasses datasets featuring long responses with complex interactions, either in text or multimodal context. During SFT, these two parts of data are concatenated and sequentially fed into the model. For MiniCPM-Llama3-V 2.5, we integrate 2M data from the recent Cauldron dataset[9] for multimodal knowledge augmentation, and 90K multilingual data over 36 languages for boosting the multilingual conversation capability.

## Alignment

MLLMs are typically prone to hallucination problems, generating responses that are not factually grounded in the input image[19]. The issue greatly limits the wide application of MLLMs, especially in high-stakes scenarios, such as autonomous driving and assistance for visually impaired groups. To address the hallucination problem, we employ the recent RLAIF-V[18] approach, where the key is to obtain scalable high-quality feedback from open-source models for preference learning[21].

**Response generation.** The first step of RLAIF-V is to generate multiple responses for a given instruction using the policy model. Specifically, given a model **M** waiting for alignment, we sample 10 responses $Y = \{y_1, y_2, \cdots, y_n\}$ from **M** using sampling decoding with high temperatures. There are several benefits of using the policy model **M** for response generation: (1) Feedback collection and learning can better focus on trustworthiness, since different text styles from multiple MLLMs are avoided. (2) Feedback learning is more efficient since preference is directly collected on the distribution of the policy model.

**Feedback collection.** Collecting high-quality feedback from open-source MLLMs can be challenging due to their typically weaker capabilities compared with proprietary models. To address the issue, RLAIF-V uses a divide-and-conquer strategy for response scoring. Specifically, each response $y_i$ is divided into atomic claims $C_i = \{c_1, c_2, ..., c_m\}$ using Llama-3 8B, where the correctness of atomic claims is much easier to evaluate. Then, we verify the claims by converting each claim to a yes/no question and employing an open-source MLLM to score each claim. In practice, we adopt OmniLMM 12B for MiniCPM-V 2.0 scoring and LLaVA-NeXT-Yi 34B for MiniCPM-Llama3-V 2.5 scoring. The final score $s_i$ of the response $y_i$ is given by $-n_{rej}$, where $n_{rej}$ is the number of invalid atomic claims.

**Direct preference optimization.** After collecting the high-quality AI feedback, we perform preference learning via DPO method. The DPO algorithm requires training on preference pairs, where one sample $y_w$ is preferred to the other one $y_l$. To compose the preference dataset, we randomly sample pairs from each response set $Y = \{y_1, y_2, ..., y_n\}$, and determine $(y_w, y_l)$ based on their relative scores. Finally, we construct a preference dataset consisting of 6K preference pairs from 3K unique images for preference learning.

## Data availability

The datasets used for training are described in detail in the Methods section. While most of the training data are publicly available, a small portion originates from proprietary datasets licensed from a commercial provider and cannot be shared due to legal and contractual restrictions. These restricted datasets were used exclusively for model training and do not affect the reproducibility of the core findings presented in this study. Source data are provided with this paper.

## Code availability

Code for training and evaluating our model is publicly available on GitHub at https://github.com/OpenBMB/MiniCPM-o, and has been archived at https://doi.org/10.5281/zenodo.15525638.

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

## Acknowledgements

Z.L. and M.S. are supported by the Research Project 2025QGS16007. Y.Y. is supported by the Shanghai Qi Zhi Institute Innovation Program SQZ202410.

## Author contributions

Yuan Yao initiated and led the research project, and designed the models and experiments. Tianyu Yu, Chongyi Wang, Junbo Cui, Hongji Zhu, Haoyu Li, Zhihui He, Haoye Zhang, Zhi Zheng, Jie Zhou and Jie Cai contributed to the experiments. Chi Chen, Ao Zhang and Yuan Yao wrote the paper. Tianchi Cai, Weilin Zhao, Qianyu Chen, Ronghua Zhou and Zhensheng Zou contributed to the open-source work. Shengding Hu, Xu Han, Guoyang Zeng, Dahai Li, Zhiyuan Liu and Maosong Sun provided valuable suggestions and proofread the paper.

## Competing interests

The authors declare no competing interests.

## Additional information

**Peer review information** : *Nature Communications* thanks the anonymous, reviewer(s) for their contribution to the peer review of this work. A peer review file is available.

