## [Transparent Peer Review file · Nature Communications]

Efficient GPT-4V Level Multimodal Large Language Model for Deployment on Edge Devices

Corresponding Author: Professor Zhiyuan Liu

Version 0:

Reviewer comments:

Reviewer #1

(Remarks to the Author)

This work presents a low-cost Multimodal Large Language Model capable of understanding vision tasks. The article is well-written, and the explanations are thorough. The ideas presented are interesting, and the contributions are significant, especially considering how resource-intensive and costly current LLMs are.

However, there are several questions and comments regarding the article and its topic:

1. How effective is the model at recognizing handwriting and artistic documents, such as advertisements or graffiti?
2. Recently, DeepSeek released models that are quite affordable and can run on Apple Silicon. How does your model compare with the latest DeepSeek models?
3. On page 7, one of the figures is not correctly referenced.
4. Regarding edge computing, single-board computers are becoming more popular due to their portability and ease of use. Would it be possible to include Nvidia Jetson, single-board computers, or any of their variants in the throughput and latency benchmarks? How long the maximum latency should be for a MLLMs to be considered sufficient?

(Remarks on code availability)

The code is well-documented and can run well on our machine.

Reviewer #2

(Remarks to the Author)

The paper addresses the challenge of reducing the hardware requirements of Multimodal Large Language Models. The proposed approach primarily focuses on handling high-resolution images and implementing a multi-step training pipeline.

On the one hand, the authors conducted an extensive set of experiments and comparisons, providing a solid experimental setup. On the other hand, the proposed approach appears modest, as it seems more like an optimization of certain critical steps rather than a truly novel method.

Adaptive Visual Encoding employs a striding strategy followed by compression to reduce the number of tokens. However, the current version does not clearly outline the computational gains compared to the standard approach. In other words, while the authors effectively explain the procedure, a more detailed comparison of its computational cost relative to the standard method is needed.

Additionally, for both the high-resolution and training strategies, an ablation study would be valuable to compare the proposed approach with naïve baselines. For example, what would be the effect of simply downsampling the input resolution? What is the actual gain introduced by the proposed training pipeline compared to standard training configurations? Providing such comparisons would help better understand the effectiveness and necessity of the proposed methods.

For the section “Efficient Deployment of MiniCPM-V on Edge Devices,” I have the same observation as for the rest of the paper. The authors have explored various existing toolkits that enabled them to achieve highly effective implementations. However, the outcome appears to be more focused on optimization rather than a genuine research exploration.

Minors:

Figure 3 is not referenced in the text.

Figure 4 lacks of the x-axis units.

Typo: "latency of 64.2s and a text decoding speed of 1.3 tokens/s (as depicted in Fig. ??),"

(Remarks on code availability)

Version 1:

Reviewer comments:

Reviewer #1

(Remarks to the Author)

Our comments from previous round of Review have been addressed well. We believe that the paper's contributions and novelties are significant enough to be published in this journal.

(Remarks on code availability)

The code is well structured and the instruction for running the code is very clear. As far as we are aware, we found no manipulation in the code. Lastly, we found that it performs as well as written in the manuscript.

Reviewer #2

(Remarks to the Author)

The authors satisfactorily replied to all my questions and remarks. I have no further concerns.

(Remarks on code availability)

The authors satisfactorily replied to all my questions and remarks. I have no further concerns.

General Response

Thanks for all of the reviewers’ time to write valuable and constructive comments. We have taken the suggestions seriously, supplemented additional experiments and analysis, and provided a point-by-point response here.

Response to the Reviewer #1

Q1: How effective is the model at recognizing handwriting and artistic documents, such as advertisements or graffiti?

A1: MiniCPM-V shows good performance in recognizing handwritten and artistic documents, as the test examples shown in Figure R.1. Quantitatively, OCRBench includes two related sub-tasks: Scene Text-centric VQA (containing advertisements and graffiti) and Handwritten Mathematical Expression Recognition. Table R.1 shows that MiniCPM-V outperforms concurrent models in both tasks.

Model	Size	OCRBench	STVQA	HMER
Proprietary				
Gemini Pro	-	68.0	87.0	9.0
GPT-4V (2023.11.06)	-	64.5	81.5	9.0
Open-source				
Yi-VL-6B	6.7B	29.0	52.0	2.0
Yi-VL-34B	34B	29.0	57.5	2.0
DeepSeek-VL-1.3B	1.7B	41.3	73.0	5.0
Qwen-VL-Chat	9.6B	48.8	78.5	0.0
DeepSeek-VL-7B	7.3B	43.5	75.0	6.0
CogVLM-Chat	17.4B	59.0	83.5	0.0
Phi-3-vision-128k-instruct	4.2B	63.9	79.0	0.0
MiniCPM-V 1.0	2.8B	36.6	66.0	0.0
MiniCPM-V 2.0	2.8B	60.5	85.5	0.0
MiniCPM-Llama3-V 2.5	8.5B	72.5	85.5	53.0

Table R.1. Accuracy results (%) on OCRBench and two related sub-tasks, including Scene-text VQA (STVQA) and handwritten mathematical expression recognition (HMER).

Q2: Recently, DeepSeek released models that are quite affordable and can run on Apple Silicon. How does your model compare with the latest DeepSeek models?

A2: The models recently released by DeepSeek are mostly text-only models without multimodal capabilities, such as DeepSeek-V3, DeepSeek-R1, and DeepSeek-R1-Distill-Qwen-7B. In terms of multimodal models, DeepSeek-VL2 (released in Dec. 2024) and Janus-Pro (released in Jan. 2025) are the latest versions. Given the rapid development of MLLMs, we make a fair comparison between these multimodal models and the concurrent MiniCPM-o 2.6, the latest version of the MiniCPM-V series released in Jan. 2025. MiniCPM-o 2.6 adopts the same vision-language architecture with MiniCPM-Llama3-V 2.5, while optimizing LLM initialization and data construction.

We perform evaluation from two perspectives, including performance and efficiency. For performance evaluation, we adopt OpenCompass 2.0, a comprehensive collection of 8 popular benchmarks, to evaluate the general multimodal capabilities. For efficiency, we compare model initialization memory (Init. Mem.), time to first token (TTFT), runtime memory (Run. Mem.), and decoding throughput. Table R.2 shows the evaluation results on a single A100 80GB GPU with batch-size = 1. We observe that MiniCPM-V series

models outperform the DeepSeek-VL series and Janus-Pro in achieving **stronger capabilities with comparable or better inference efficiency**. Note that DeepSeek-VL series are MoE (Mixture of Experts) models, which are characterized by a large number of overall parameters, sparse activation, and MLA (Multi-head Latent Attention). These designs are optimized for cloud-based high-concurrency scenarios but are not well-suited for edge-side deployment.

Q3: On page 7, one of the figures is not correctly referenced.

A3: Thanks for the suggestion! The reference has been corrected in the paper.

Q4: Regarding edge computing, single-board computers are becoming more popular due to their portability and ease of use. Would it be possible to include Nvidia Jetson, single-board computers, or any of their variants in the throughput and latency benchmarks? How long the maximum latency should be for a MLLMs to be considered sufficient?

A4: We evaluate MiniCPM-Llama3-V 2.5 on the Jetson AGX Orin 32GB, and report the results in Figure R.2 (also updated in Figure 6f). We observe that MiniCPM-V performs well on Jetson AGX Orin, showing its promising potential in edge computing. In terms of maximum acceptable latency, based on our testing experience, a delay of 3 seconds is the maximum time for good user experience. Delays longer than 3s will be perceived as a noticeable lag, while delays exceeding 10 seconds will cause users to lose focus and shift their attention elsewhere. As for throughput, the average human reading speed is approximately 5 tokens per second. A throughput above 8 tokens/s is considered smooth output, while a rate above 15 tokens/s is perceived as fast output.

Response to the Reviewer #2

Q1: Adaptive Visual Encoding employs a striding strategy followed by compression to reduce the number of tokens. However, the current version does not clearly outline the computational gains compared to the standard approach. In other words, while the authors effectively explain the procedure, a more detailed comparison of its computational cost relative to the standard method is needed.

A1: Thanks for the suggestion. We provide a comparison of the computational costs between our proposed adaptive visual encoding method and the vanilla visual encoding method (i.e., visual features of the original image from ViT are directly projected and input into the LLM, as in LLaVA-1.5 [1]). From results in Figure R.3 (also added in Figure 2a), we can see that compared with the standard method, adaptive visual encoding largely reduces both FLOPs and GPU memory usage in both ViT and LLM components for high-resolution images. The reason is that, compared with the standard method, image slicing prevents the quadratic computation growth of ViT, and the compression layer largely reduces the number of visual tokens to LLMs.

Q2: Additionally, for both the high-resolution and training strategies, an ablation study would be valuable to compare the proposed approach with naive baselines. For example, what would be the effect of simply downsampling the

Figure R.1. Qualitative results of MiniCPM-V on handwriting and artistic documents (advertisements and graffiti images).

Model	Params	OpenCompass 2.0	Init. Mem.	TTFT (img size=1344)	Run. Mem. (img size=1344)	Decoding Throughput
DeepSeek-VL2-Small	16.1B (2.8B activated)	64.5	37G	360ms	39G	19.5 tokens/s
DeepSeek-VL2	27.5B (4.5B activated)	66.4	68G	550ms	72G	14.2 tokens/s
Janus-Pro	7.4B	50.2	14G	270ms	16G	48.8 tokens/s
MiniCPM-o 2.6	8.6B	70.2	16G	300ms	17G	40.0 tokens/s

Table R.2. Comparison with DeepSeek-VL and Janus-Pro. For efficiency, we compare model initialization memory (Init. Mem.), time to first token (TTFT) and runtime memory (Run. Mem.), as well as decoding throughput.

Figure R.2. Results on various edge devices, with added performance for the NVIDIA Jetson.

Model	OpenCompass 2.0	MMMU (val)	MMVet	MMBv1.1 EN (val)	OCRBench	ChartQA (test)	MME	TextVQA (val)	DocVQA (test)
MiniCPM-Llama3-V 2.5	54.1	43.3	46.7	69.6	697	65.4	1922	72.5	79.5
w/o multi-stage training	50.1	42.9	44.9	67.5	549	57.6	1544	70.1	64.7
w/o high-resolution perception	48.5	37.9	37.8	64.2	580	51.2	1701	65.1	58.4

Table R.3. Multi-stage training and high-resolution perception ablation results on a subset (10%) of full training data.

Figure R.3. Comparison of computational costs (including FLOPs and GPU memory usage) between vanilla visual encoding and the adaptive visual encoding (ours) across different image sizes.

input resolution? What is the actual gain introduced by the proposed training pipeline compared to standard training configurations? Providing such comparisons would help better understand the effectiveness and necessity of the proposed methods.

A2: We perform an ablation study on high-resolution perception and multi-stage training pipeline. To investigate the effectiveness of high-resolution perception, we follow the standard method in LLaVA-1.5 [1] to downsample high-resolution images into low-resolution versions (i.e., 448×448) in both training and inference. To ablate multi-stage training pipeline, we follow the standard two-stage training (i.e., pretraining and instruction tuning) in LLaVA-1.5 [1]. Due to the high computational costs of model training on full data, we perform ablation on a subset of full training data by randomly sampling 10% data from each dataset, resulting in 70M data in total, which is sufficient to validate a frontier MLLM. From the experimental results in Table R.3, we can see that both high-resolution perception and multi-stage training pipeline contribute to the final performance. The reason is that high-resolution perception is crucial for MLLMs to perceive fine-grained visual details, especially for OCR-related tasks, and multi-stage training can better fit and exploit training data of different forms and qualities.

Q3: For the section “Efficient Deployment of MiniCPM-V on Edge Devices,” I have the same observation as for the rest of the paper. The authors have explored various existing toolkits that enabled them to achieve highly effective implementations. However, the outcome appears to be more focused on optimization rather than a genuine research exploration.

A3: We would like to highlight that the primary focus and contribution of this work lie in the innovative aspects of the proposed model, including adaptive visual encoding mechanism and progressive multimodal learning strategy, etc. Additionally, while there is a growing consensus within the community regarding the potential of edge-side MLLMs, there remains a notable lack of systematic experimental reports about their deployment on edge devices. We believe this empirical study could offer benefits to the community in the following ways: (1) For the broader community, it can provide a clearer understanding of the current

development level in this field, particularly regarding how MLLMs perform when deployed on edge devices. (2) For practitioners, it can offer insights into design choices and best practices for deploying MLLMs on edge devices, which could help promote applications of edge-side MLLMs. (3) For researchers, it could provide feedback on the actual performance of MLLMs on edge devices, which may inspire the design of more efficient model architectures in future.

We sincerely view this section as a beneficial supplement to the primary work of this paper. We hope that it will contribute to the ongoing development and adoption of edge-side MLLMs within the community. Thank you for your thoughtful consideration.

Q4: Figure 3 is not referenced in the text. Figure 4 lacks of the x-axis units. Typo: “latency of 64.2s and a text decoding speed of 1.3 tokens/s (as depicted in Fig. ??),”

A4: Thanks for the suggestion. All these typos have been fixed in the revised version.

References

- [1] H. Liu, C. Li, Y. Li, and Y. J. Lee, “Improved baselines with visual instruction tuning,” in *Proceedings of the IEEE/CVF Conference on Computer Vision and Pattern Recognition*, pp. 26296–26306, 2024.

Point-by-Point Response to Reviewers' Comments

Reviewer #1

Remarks to the Author:

We thank the reviewer for the positive feedback and are pleased to hear that the contributions and novelties of our work are considered significant enough for publication. We appreciate the acknowledgement that our revisions have addressed the prior concerns effectively.

Remarks on Code Availability:

We are grateful for the reviewer's recognition of the code quality and clarity of instructions.

Reviewer #2

Remarks to the Author:

We sincerely thank the reviewer for carefully evaluating our revised manuscript and for the positive assessment. We are glad to know that all comments and questions have been satisfactorily addressed.

Remarks on Code Availability:

We appreciate the reviewer's confirmation that all issues have been fully addressed. Thank you for the supportive feedback.